# Clinical and Radiographic Outcomes of Inversed Restricted Kinematic Alignment Total Knee Arthroplasty by Asia Specific (Huang’s) Phenotypes, a Prospective Pilot Study

**DOI:** 10.3390/jcm12062110

**Published:** 2023-03-08

**Authors:** Shang-Lin Hsieh, Tsung-Li Lin, Chih-Hung Hung, Yi-Chin Fong, Hsien-Te Chen, Chun-Hao Tsai

**Affiliations:** 1Department of Orthopedics, China Medical University Hospital, Taichung 404, Taiwan; 2Graduate Institute of Biomedical Sciences, China Medical University, Taichung 404, Taiwan; 3Department of Sports Medicine, College of Health Care, China Medical University, Taichung 404, Taiwan; 4Department of Orthopedics, China Medical University Beigang Hospital, Yunlin 651, Taiwan

**Keywords:** inverse restricted kinematic alignment (irKA), total knee arthroplasty, knee alignment, patient satisfaction, FJS-12, WOMAC

## Abstract

Inverse restricted kinematic alignment (irKA) was modified from restricted kinematic alignment for total knee arthroplasty (TKA). This prospective single-center study aimed to evaluate the outcomes of irKA-TKA on all knee subtypes classified by Asia specific (Huang’s) phenotypes. A total of 96 knees that underwent irKA-TKA at one hospital between January 2018 and June 2020 were included, with 15 knees classified in Type 1, nine in Type 2, 15 in Type 3, 47 in Type 4, and 10 in Type 5 by Huang’s phenotypes. Outcomes were knee alignment measures and patient-reported satisfaction evaluated by the Western Ontario and McMaster Universities Arthritis Index (WOMAC) and traditional Chinese version of the Forgotten Joint Score-12 (FJS-12). Follow-up was one year. Type 4 knee was most significantly corrected in all angles by irKA-TKA, followed by Type 2 and 3 knees. Type 5 and 1 knee were only significantly corrected in some angles. The correlation between FJS-12 and WOMAC was good at 6 months (Pearson correlation coefficient (r) = 0.74) and moderate at 6 weeks, 3 months, and 12 months during follow-up (r = 0.37~0.47). FJS-12 and WOMAC displayed comparable hip–knee–ankle angle cut-off value (4.71° vs. 6.20°), sensitivity (70.49% vs. 67.19%), specificity (84.00% vs. 85.71%), and Youden index (54.49% vs. 52.90%) in prediction of good prognosis. In conclusion, irKA-TKA corrects knee alignment in all knee types with increasing satisfaction for one-year follow-up. Knees with presurgical varus deformity are most recommended for irKA-TKA. Both presurgical scores of the traditional Chinese version of FJS-12 and WOMAC predict the prognosis of irKA-TKA.

## 1. Introduction

Osteoarthritis is a debilitating condition that causes disability, psychological stress, and poor quality of life. Total knee arthroplasty (TKA) is a frequently used orthopedic procedure to treat severe osteoarthritis and improve a patient’s mobility and well-being. Singh et al. [1] used data from both of the 2000–2014 USA National Inpatient Sample and Census Bureau and predicted the annual incidence of primary TKA can be 1.065 and 3.416 million at 2020 and 2040, respectively. The main goal of TKA is rebuilding a stable knee with a neutrally aligned lower limb for satisfied outcomes and extending the life-cycle of implants [2,3]. Various knee alignment procedures are known. Mechanical alignment (MA) positions include both femoral and tibial components perpendicular to the mechanical axis of each bone, making the hip–knee–ankle angle (HKAA) be 0° or 180° [4,5]. Kinematic alignment (KA), which places emphasis on the natural knee alignment and soft tissue balance, was proposed to provide better satisfaction than MA [6]. Restricted kinematic alignment (rKA) tolerated the results of tibial and femoral cuts within 5° of the mechanical axis and the overall alignment within ±3° of neutral [7,8,9]. This allows an expanded safe zone of alignment and higher degree of satisfaction.

Inverse restricted kinematic alignment (irKA), a tibia-first and gap balancing version of rKA, was developed for excellency [10,11]. A previous study reported that irKA-TKA granted comparable clinical outcomes in varus, neutral, and valgus knees as MA-TKA [12]. They also showed that patients who received irKA-TKA had better satisfaction than MA-TKA, evaluated by the Oxford Knee Score within 12-month follow-up. This indicates that irKA-TKA can be a good option for treating osteoarthritis. Due to native varus in the medial proximal tibial angle (MPTA) in Asian, the Asia specific phenotype, Huang’s phenotypes, grouped knees into five phenotypes based on HKAA, lateral distal femoral angle (LDFA), and MPTA [13]. The purpose of this study was to determine the clinical outcome of irKA-TKA in all knees separated by Huang’s phenotypes. The null hypothesis was irKA-TKA and it yielded similar outcomes and satisfaction scores among all phenotypes. The finding could improve the clinical outcome while restoring the physiological joint line obliquity in TKA.

## 2. Methods

### 2.1. Patient Selection

Patients older than 60 years old and who received inversed restricted kinematic alignment (irKA) total knee arthroplasty (TKA) between January 2018 and June 2020 in the China Medical University Hospital were included for this study. Patients with revision surgery, previous trauma around the knee, knee collateral ligament injury, pre-knee malignancy, and losing follow-up within 1 year were excluded.

### 2.2. Angle Determination

All the patients had long-leg standing X-rays radiographs according to the protocol suggested by Ramadier et al. [14] and Saragaglia et al. [15] pre-operatively and within three months post-operatively. Four different angles including hip–knee–ankle angle (HKAA), knee alignment angle (KAA), lateral distal femoral angle (LDFA), and medial proximal tibial angle (MPTA) were measured and studied in this study.

The frontal plane was first accurately determined by looking for a true lateral view of the knee. This was done by rotating the posture until the posterior margins of the condyles were superimposed. While maintaining the X-ray source perpendicular to the frontal plane, the image intensifier was turned 90° around the knee to obtain an accurate long-leg AP standing view. The footprint was drawn on cardboard to reproduce the same orientation of the lower leg pre- and postoperatively. The same view can be reached by placing the foot on the cardboard afterward. The HKAA, KAA, LDFA, and MPTA could then be measured from the long-leg film.

This protocol was used for all the patients to avoid any errors in the measurements related to the rotation of the lower leg. The measurements were done with a ruler and a manual goniometer by one radiologist and one orthopedic practitioner independently.

### 2.3. Surgery

On the day of the surgery, a liquid or semi-liquid diet was provided. To reduce bleeding during the surgery, one gram of tranexamic acid (Daiichi Sankyo Taiwan Ltd., Taipei, Taiwan) dissolved in 100 mL of 0.9% NaCl was infused intravenously thirty minutes before the operation.

All patients were implanted with the Zimmer^®^ NexGen^®^ LPS Fixed Knee system (Zimmer Biomet, Warsaw, Indiana, USA). The tibia resection was conducted first to restore the pre-arthritic MPTA within the safe zone. The restricted safe zone was defined as 86° to 93° for recreation of both the LDFA and the MPTA, and −5° varus to +4° valgus for the final HKAA and KAA. A medial parapatellar arthrotomy approach was chosen. The tibial cut was aimed to reconstruct the native tibial joint line obliquity. Then, after tibial cut (Figure 1A), two laminar spreaders were used to separate the tibia and femur and provide tension on the tendon (Figure 1B). A 10 mm block-spreader was used to draw the line of resection at either posterior or distal side of the femur (Figure 1D–G). Femoral rotation was initially planned parallel to the resection line at the posterior femoral bone but adjusted if the tibial resection had to be reduced to fall within the safe zone and according to the trans-epicondylar line (Figure 1F). The IM external-rotation guide-plate was aligned with the line of femoral rotation and used to determine the angle of distal resection (Figure 1H). The anterior cut was conducted (Figure 1I). The angle of the distal cut was determined based on the drawn line (Figure 1J). Then, resection was conducted (Figure 1K).

To prevent postoperative limb swelling and pain, a tourniquet was not used during the surgery. After joint capsule closure, 60 mL of 0.9% NaCl containing one gram of tranexamic acid was infused in the articular cavity again to reduce bleeding.

### 2.4. Postsurgical Management

Several procedures were conducted to enhance recovery after irKA-TKA surgery. Intermittent cold pressure dressing and compression on the incision were applied to stop bleeding after the surgery, and a drainage tube was not indwelled. Elastic bandages were applied to promote venous return after the operation. The patients were enjoined to lift both lower limbs to reduce the swelling of the affected limb.

### 2.5. Rehabilitation

Joint mobilization and crutches supporting immediate weight bearing were encouraged for rehabilitation. Physiotherapist-guided full joint movement, including active flexion and extension, was performed from day 1. Patients stayed in the hospital for 3 nights on average. Daily physiotherapy was continued after discharge. Patients were advised to use crutches during the first two weeks. For prevention and protection, all patients received Aspirin 100 mg (Yung Shin Pharmaceutical Industrial Co., Ltd., Taichung, Taiwan) for 4 weeks after surgery. The first follow-up appointment was 6 weeks after surgery.

### 2.6. Outcome Measures

The knee orientation was measured in terms of HKAA, KAA, LDFA, and MPTA to determine whether the native limb and knee alignment was restored to the natural physiological state after irKA-TKA. The HKAA, KAA, LDFA, and MPTA were measured from the films of the X-ray radiography taken at 6 weeks follow-up.

Patients were asked to evaluate the pain or discomfort in their daily life before and after irKA-TKA surgery by the Western Ontario and McMaster Universities Arthritis Index (WOMAC) and the traditional Chinese version of Forgotten Joint Score-12 (FJS-12). WOMAC and FJS-12 were conducted by questionnaire to quantify the severity of knee pain with daily activities at 6 weeks, 3 months, 6 months, and 12 months after surgery. The WOMAC was also reported before surgery for comparison.

### 2.7. Ethnical Claim

This was a longitudinal prospective cohort study of a single surgeon series registered at clinicaltrials.gov (NCT05685706) and performed in accordance with the declaration of Helsinki. The study protocol was reviewed and approved by the China Medical University and Hospital Research Ethics Committee (CMUH110-REC2-030). Written informed consent was signed by all the participating subjects.

## 3. Statistical Analysis

As for the comparison between five types of phenotypes, continuous data with normal distribution were presented as mean ± standard deviation (SD) using Student’s test and data without normal distribution were presented as the median and interquartile range (IQR: 25th—75th percentile) using the Wilcoxon rank sum test. Categorical data were presented as *n* (%) using Fisher’s exact test. The distributions of LDFA/MPTA before and after operation were presented in frequency histograms. The changes of HKAA, KAA, LDFA, and MPTA before and after operation were examined by Wilcoxon sign-rank tests. Trend tests for the improvement in the FJS-12 and WOMAC scores were calculated from simple linear regression models with the FJS12/WOMAC scores as a response variable and the order of time course as a predictor. The consistency of FJS-12 and WIMAC for each time point was performed by Spearman correlation analysis. The sensitivity and specificity of predicted results were calculated to make a receiver operating characteristic (ROC) curve. The cutoff point of HKAA before operation was decided by the maximum value of the Youden index. Each area under the curve (AUC) was calculated, with a higher AUC indicating higher predictive performance. We extracted the maximum AUC to define the prognosis by the FJS-12 and WOMAC score and HKAA before operation as the independent variable using univariate logistic regression. If a patient had a prognosis FJS-12 score ≥ 72.9 or WOMAC score ≥ 82, then the patient will be considered to have good prognosis. Using the FJS-12 ≥ 72.9 or WOMAC score ≥ 82 as a good prognosis, we identified the cutoff point of HKAA before operation to predict the prognosis of our patient. All statistical analyses were performed by using SAS software (version 9.4 for Windows; SAS Institute, Cary, NC, USA). A two-sided *p* < 0.05 was established as statistically significant.

## 4. Results

In total, 76 patients with 96 knees were included in this study. It included 60 females (75 knees) and 16 males (21 knees). The mean age was 70.4 ± 9.7 years old. For the knee classification, 15, 9, 15, 47, and 10 knees were grouped into Type 1, 2, 3, 4, and 5, respectively. The baseline information of the study population is presented in Table 1. The median surgical time of all the patients was 108.0 min and the median blood loss was 50.0 mL.

Distributions of LDFA/MPTA before and after surgery are presented as the frequency histograms in Figure 2. Pre-operationally, 71.9% knees had MPTA < 87° and 58.3% knees had LDFA ≥ 90°, which indicated most knees in this study were varus. On the contrary, valgus knees, which is defined as LDFA < 87°, only occupy 17.7% of knees. After surgery, patients with varus knees reduced from 58.3% to 44.8% in LDFA ≥ 90° and 71.9% to 34.4% in MPTA < 87°. In the valgus condition, patients with LDFA < 87° reduced from 17.7% to 3.1%.

Table 2 shows the irKA surgery corrected coronal alignment in different generic types by classification. After irKA surgery, HKAA, KAA, and LDFA reduced (*p* < 0.001, *p* = 0.006 and *p* = 0.009) and MPTA increased (*p* < 0.001) in the whole study population. In Type 1, HKAA was the only parameter that changed significantly (median: 2.6 to 1.5, *p* = 0.002). In Type 2 and 3, HKAA in each group reduced significantly (*p* = 0.027 and 0.003) and LDFA and MPTA significantly increased. In Type 4, HKAA, KAA, and LDFA significantly reduced (*p* < 0.001, 0.014, and <0.001, respectively) and MPTA increased (median: 81.1° to 87.0°, *p* < 0.001). In Type 5, LDFA significantly increased (median: 85.2° to 87.6°, *p* = 0.004) and MPTA reduced (median: 90.6° to 86.4°, *p* = 0.004).

Table 3 shows the functional outcome within one year follow-up by FJS12 and WOMAC in different phenotypes. Trend tests for the improvement in FJS-12 and WOMAC scores were calculated from simple linear regression models with the FJS12/WOMAC scores as a response variable and the order of time course as a predictor. FJS-12 had a significant increasing trend from the 6th week to 12th month in neutral and varus type (both *p* < 0.001) and the significance of valgus was not found (*p* = 0.068) because of the reduced mean score from the 3th month to 6th month (83.33 to 63.54). WOMAC had a significant increasing trend from pre-op to the 12th month in each type (all *p* < 0.001). Neutral and varus knees had the largest increase from pre-op to the 6th week, specifically. Comparing FJS-12 and WOMAC at each follow-up time, the significant positive correlations were found at the 6th month (r = 0.74, *p* < 0.001) and 12th month (r = 0.47, *p* = 0.005).

The receiver operating characteristic (ROC) curve of HKAA is shown in Figure 3. The cut was at 72.9 for the FJS-12 score and at 82 for the WOMAC score. By setting the FJS-12 score higher than 72.9 as the indicator of good prognosis, we found that a cut-off HKAA before operation of 4.71° had the largest Youden value of 54.49%. The sensitivity and specificity at that point were 70.49% and 84.00%, respectively. The ROC curve of the model analysis showed the area under the curve was 72.87% (Figure 3A). Again, by defining the indicator of good prognosis of the WOMAC score as higher than 82, we identified the cutoff point of HKAA before operation to predict the prognosis of our patient. The cutoff in HKAA was 6.20° at the largest Youden value of 52.90%. The sensitivity, specificity, and AUC of the model were 67.19%, 85.71%, and 72.77%, respectively (Figure 3B).

## 5. Discussion

Our study indicated that irKA-TKA restored the pre-arthritic MPTA, LDFA, and HKAA in all knees classified by Huang’s phenotypes. The FJS-12 and WOMAC showed a similar trend of change with increasing satisfaction within the 12 month follow-up. The correlations between FJS-12 and WOMAC was good at 6 months (r = 0.74) and moderate at 6 weeks, 3 months, and 12 months (r = 0.37, 0.40, and 0.47, respectively) as a measure of the Pearson correlation coefficient.

irKA is a tibia-first and gap balancing version of rKA, which is derived from KA and MA. rKA increased the ratio of satisfaction from 81% patients in MA to 93% [16]. Additionally, because of method of resection, patients receiving irKA are expected to have better satisfaction than those with rKA. In the irKA, medial and lateral tibia are resected with equal thickness. This makes the new tibial slope parallel to the natural slope of the medial tibia and restores the joint line obliquity. To accommodate the femoral implant and restore the joint line height, the medial distal femur is resected with equal thickness of the implant. The resections are necessary and minimal in irKA. On the contrary, equal thickness of bones are resected at the medial and lateral femur in rKA [17,18]. To balance the knee and compensate the femoral resection, a great amount of posterior tibial resection is necessary to restore the joint line obliquity. This may induce patellar instability [19,20] and lead to imprecise resection, especially in patients with severe constitutional coronal plane deformity [21]. Another method is called adjusted MA, which considers the natural ligamentous tension of the knee joint to improve the satisfaction of MA [22]. Winnock de Grave et al. [12] compared the outcome between irKA and adjusted MA, in which knees with preoperative varus deformity were reported to achieve a significantly better outcome and satisfaction.

Another study determined the threshold of the FJS-12 for detecting the patient acceptable symptom state (PASS) following primary TKA [23]. They reported that the threshold score of the FJS-12, which maximized the sensitivity and specificity for detecting a PASS, was 33.3 (AUC = 0.78, 95% CI (confidence interval) [0.74, 0.83]). The AUC we predicted for good prognosis was 72.77% (Figure 3B), which is close to their prediction. Although the philosophy of alignment was not mentioned in their study, it indicated that FJS-12 had the potential to assess the successful achievement of TKA. The traditional Chinese version of the FJS-12 questionnaire was used to evaluate the outcome of irKA-TKA along with the widely accepted WOMAC. Since the official language is Mandarin and the traditional Chinese characteristics are used in Taiwan, a traditional Chinese version of FJS-12 can be understood by most patients directly. This avoids any bias induced by translation and interpretation. WOMAC is widely used in the evaluation of hip and knee osteoarthritis [24]. However, FJS-12 is a relatively new index to assess outcomes after joint arthroplasty [25]. The two indices both focus on the joint pain and stiffness experienced by the patient, as well as their ability to perform daily activities. The main difference between these two indices is the specific symptoms they focus on. Although both indices have been used to evaluate the outcome of TKA in other reports [26,27], the association between the two indices was not mentioned. We reported that WOMAC and FJS-12 had good correlation at 6 months and moderate correlation at 6 weeks, 3 months, and 12 months measured by the Pearson correlation coefficient. Besides, the HKAA cut-off value, sensitivity, specificity, and Youden index of good prognosis is similar between FJS-12 and WOMAC. Hence, the traditional Chinese version of FJS-12 seems to be an applicable questionnaire for evaluating the outcome of TKA.

In this study, we predicted that HKAA higher than 6.20° or 4.71° had a good prognosis by WOMAC and FJS-12, respectively. Another study reported that patients with preoperative HKAA more than 10 degrees varus had a better outcome when receiving KA than MA [28]. Thus, irKA-TKA might generate a good prognosis in some patients who are not recommended to receive KA. Our results also showed that although the physiological joint line obliquity was restored by irKA-TKA in all phenotypes, the effect of irKA-TKA was different among different Huang’s phenotypes. Knees in Type 4 (General vara:Tibia vara + Femur Vara) showed significant improvement in all measured angles, including HKAA, KAA, LDFA, and MPTA. This indicated that patients with Type 4 knee may be the best candidates for irKA-TKA. Patients with Type 1 and 5 knees lost less blood than other phenotypes during the operation even though the knee of all phenotypes was restored to the natural physiological state, suggesting that irKA-TKA has less damage to the surrounding tissue in Type 1 and 5 knees.

The purpose of TKA is restoring the native harmony of the knee. Three interdependent elements, including morphology, soft-tissue balance, and alignment, should be considered at the same time to reach this goal [29,30]. As shown in Figure 4, MA, which solely focuses on alignment and ignores the other two elements, has a high ratio of dissatisfaction. Other alignment strategies, such as adjusted mechanical alignment (aMA), anatomical alignment (AA), KA, rKA, and irKA, were developed to move toward the harmony. Among these alignment philosophies, KA and its derivatives try to minimize the change in the soft-tissue balance. Additionally, irKA is not only a soft-tissue respecting strategy, but also provides a chance for intraoperative adjustment [29]. In our study, the HKAA cut-off value was determined from WOMAC and FJS-12 in the knee of Asian-specific (Huang’s) phenotypes. By combining the soft-tissue respected nature of irKA and the predicted HKAA cut-off value, patients with HKAA less than varus 6.20° through combined tension gap-balance and measured resection of irKA should reduce the need of soft-tissue release to a minimum.

## 6. Limitation

There are some limitations in this study. First, the sample cohort was small. This made it difficult to generalize the finding. Second, the timeframe of follow-up was only 2 years. The conclusion may be challenged in long-term conditions. Third, there was no control group in our study. A comparison between mechanical, kinematic, or anatomic alignment in TKA would create an observation with more significance. Fourth, this was a prospective study based on a single surgeon. The protocol, custom, and facility may affect the results unintentionally.

## 7. Conclusions

irKA-TKA correct knee alignment in all knee types with increasing satisfaction during one-year follow-up. The traditional Chinese version of FJS-12 was moderate or well correlated with WOMAC during follow-up. The presurgical scores of both FJS-12 and WOMAC can predict prognosis after irKA-TKA. Knees with preoperative varus deformity are most recommended for irKA-TKA.

## Figures and Tables

**Figure 1 jcm-12-02110-f001:**
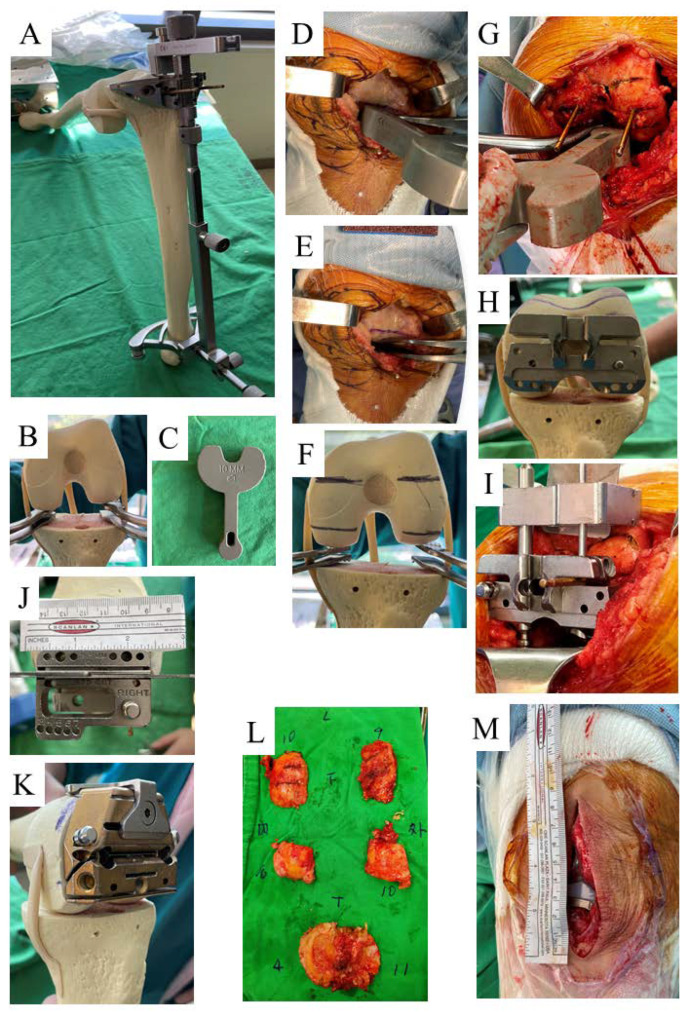
The procedure of inverse restricted KA (irKA)-TKA is illustrated. (**A**) Tibia cut was conducted first. (**B**) Laminar spreaders were used to provide tension. (**C**) A 10 mm block-spreader was used for assisting irKA. (**D**,**E**) The posterior cut was planned. (**F**) Femoral rotation was planned parallel to the resection line at the posterior femoral bone. (**G**) A 10 mm block-spreader was inserted and a parallel line against the epicondylar line was drawn. Two pins were anchored for chamfer. (**H**) The IM external-rotation guide-plate was positioned by aligning the line of femoral rotation. (**I**) Anterior cut was conducted. (**J**) Determine the cutting angle based on drawn line. (**K**) Distal cut preparation. (**L**) Resected bones were shown. (**M**) The size of wound.

**Figure 2 jcm-12-02110-f002:**
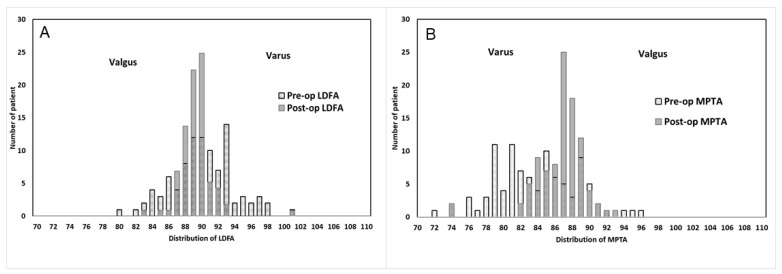
The pre-op and post-op (**A**) LDFA and (**B**) MPTA distribution. LDFA = lateral distal femoral angle, MPTA = medial proximal tibial angle.

**Figure 3 jcm-12-02110-f003:**
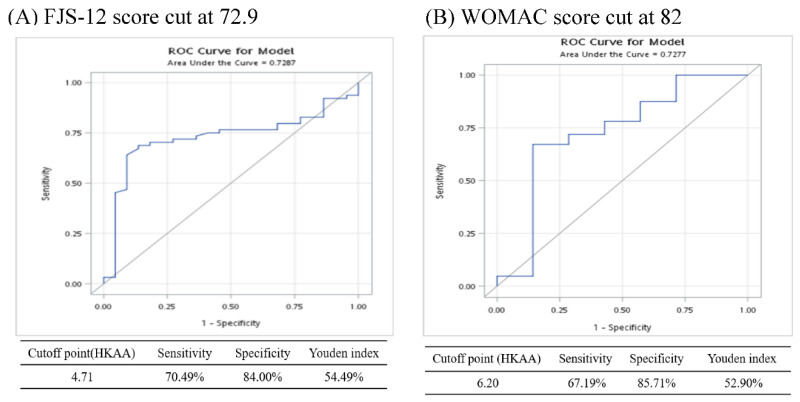
The ROC curve of (**A**) HKAA for FJS-12 score cut at 72.9 and (**B**) HKAA for WOMAC score cut at 82.

**Figure 4 jcm-12-02110-f004:**
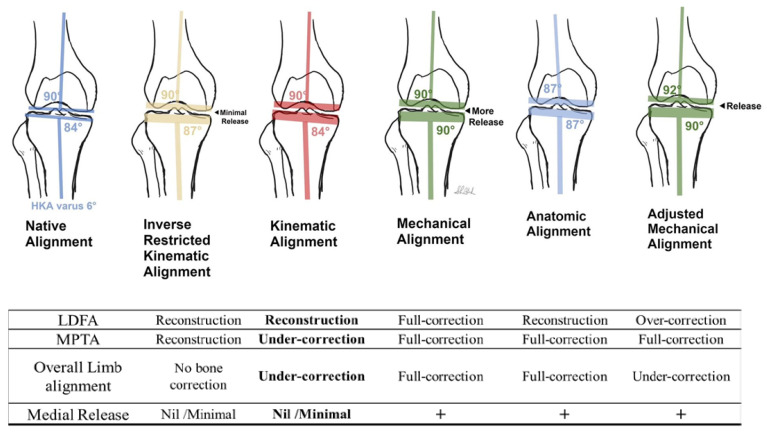
Comparison of different techniques for TKA coronal aligning. From left to right, native knee, inverse restricted KA (irKA), kinematic alignment (KA), mechanical alignment (MA), anatomical alignment (AA), adjusted mechanical alignment (aMA). The irKA technique restores alignment through combined tension gap-balanced and measured resection technique based on native tibia vara (within 6.2°HKA), reducing the need of soft-tissue release. “+”, means with medial release.

**Table 1 jcm-12-02110-t001:** Base information of study population and comparison between five types of phenotypes by Huang’s classification.

Variable	Total	Type 1 (*n* = 15)	Type 2 (*n* = 9)	Type 3 (*n* = 15)	Type 4 (*n* = 47)	Type 5 (*n* = 10)	*p*-Value
Age, years	70.4 (± 9.7)	70.0 ± 8.0	73.9 ± 6.9	71.9 ± 10.6	70.7 ± 10.2	64.1 ± 8.7	0.215
Sex							0.0502
Male	19 (19.8%)	5 (33.3%)	1 (11.1%)	2 (13.3%)	6 (12.8%)	5 (50.0%)	
Female	77 (80.2%)	10 (66.7%)	8 (88.9%)	13 (86.7%)	41 (87.2%)	5 (50.0%)	
Surgical time, min	108.0 (96.0–121.0)	105.3 (99.0–121.0)	119 (102.5–143.0)	108.0 (97.0–114.0)	103.0 (96–118.0)	98.8 (83.0–140.0)	0.350
Blood loss, mL	50.0 (25.0–50.0)	25.0 (25.0–25.0)	50.0 (50.0–50.0)	50.0 (25.0–50.0)	50.0 (25.0–50.0)	25.0 (25.0–50.0)	<0.001
Surgical site							<0.001
Left	45 (46.9%)	9 (60.0%)	5 (55.6%)	7 (46.7%)	21 (44.7%)	3 (30.0%)	
Right	51 (53.1%)	6 (40.0%)	4 (44.4%)	8 (53.3%)	26 (55.3%)	7 (70.0%)	

Continuous data with normal distribution are presented as mean ± standard deviation and data without normal distribution are presented as median (25th—75th percentile); categorical data are presented as *n* (%). Significant results are shown in bold.

**Table 2 jcm-12-02110-t002:** Correction of irKA in coronal alignment in different phenotypes.

	Pre-Op	Post Op	*p*-Value
Total (*n* = 96) ^a^			
HKAA	6.2 (2.3–10.1)	2.1 (1.5–4.5)	**<0.001**
KAA	4.7 (3.4–7.0)	3.8 (2.1–5.1)	**0.006**
LDFA	89.9 (87.7–92.7)	89.3 (88.4–90.0)	**0.009**
MPTA	83.5 (80.3–87.0)	87.1 (85.3–88.1)	**<0.001**
Type 1 (*n* = 15)			
HKAA	2.6 (1.9–3.9)	1.5 (1.2–1.7)	**0.002**
KAA	4.4 (2.8–5.8)	3.4 (2.9–4.4)	0.169
LDFA	89.5 (88.7–89.7)	88.9 (88.2–89.3)	0.188
MPTA	89.0 (88.5–89.4)	88.5 (87.5–89.9)	0.463
Type 2 (*n* = 9)			
HKAA	2.3 (1.5–4.8)	2.0 (0.9–2.3)	**0.027**
KAA	5.2 (4.2–6.7)	3.8 (1.0–4.7)	0.164
LDFA	84.3 (84.0–86.0)	89.3 (88.2–90.5)	**0.039**
MPTA	83.6 (83.1–84.7)	88.0 (86.7–89.2)	**0.004**
Type 3 (*n* = 15)			
HKAA	4.9 (2.4–8.5)	2.1 (1.5–3.8)	**0.003**
KAA	5.0 (3.2–6.4)	3.7 (2.1–6.9)	0.561
LDFA	88.4 (87.7–89.2)	89.3 (88.7–89.7)	**0.0497**
MPTA	81.2 (79.3–84.4)	87.5 (86.7–88.1)	**<0.001**
Type 4 (*n* = 47)			
HKAA	9.8 (5.8–13.9)	3.5 (1.8–5.7)	**<0.001**
KAA	4.9 (3.8–7.6)	3.6 (2.1–5.0)	**0.014**
LDFA	92.7 (91.1–93.8)	89.9 (88.4–90.1)	**<0.001**
MPTA	81.1 (78.7–84.5)	87.0 (84.5–87.4)	**<0.001**
Type 5 (*n* = 10) ^a^			
HKAA	6.8 (3.3–8.7)	2.6 (2.2–5.0)	0.055
KAA	4.5 (−12.6–12.8)	5.1 (1.8–5.4)	0.652
LDFA	85.2 (83.7–86.1)	89.4 (87.6–90.0)	**0.004**
MPTA	90.6 (89.5–94.1)	86.4 (85.7–88.3)	**0.004**

Abbreviation: HKAA, hip–knee–ankle angle; KAA, knee alignment angle; LDFA, lateral distal femur angle; MPTA, medial proximal tibial angle. Data with normal distribution are presented as median (25th—75th percentile). Significant results are shown in bold. ^a^ There was one patient missing the post-op information of HKAA, KAA, LDFA, MPTA.

**Table 3 jcm-12-02110-t003:** Time course one year follow-up functional outcomes of FJS12 and WOMAC in different Huang’s phenotypes.

Score	Pre-Op	6th Week	3th Month	6th Month	12th Month	*p*-Value for Trend ^a^
FJS-12						
Phenotype						
Neutral (*n* =24)	-	46.7 ± 17.1	62.5 ± 21.8	61.3 ± 21.8	81.5 ± 13.1	<0.001
Valgus (*n* =9)	-	69.2 ± 5.6	83.3 ± 10.8	63.5 ± 36.8	88.8 ± 8.7	0.068
Varus (*n* = 63)	-	60.8 ± 26.1	58.4 ± 27.6	71.4 ± 17.0	83.3 ± 9.0	<0.001
WOMAC						
Phenotype						
Neutral (*n* =24)	45.7 ± 12.5	80.2 ± 12.4	87.7 ± 9.6	80.5 ± 11.6	85.0 ± 6.0	<0.001
Valgus (*n* =9)	63.2 ± 13.5	65.5 ± 1.7	90.3 ± 7.7	96.0 ± 10.0	95.2 ± 4.2	<0.001
Varus (*n* = 63)	52.3 ± 16.6	94.3 ± 2.5	81.2 ± 8.0	87.6 ± 8.6	90.1 ± 8.5	<0.001
FJS-12 and WOMAC						
r	-	0.37	0.40	0.74	0.47	
*p*-value	-	0.288	0.142	<0.001	<0.005	

Abbreviation: FJS-12, Forgotten Joint Score-12; WOMAC, Western Ontario and McMaster Universities Arthritis Index. Significant results are shown in bold. ^a^ There was significant functional improvement from 6th to 12th month.

## Data Availability

The data are available by requesting.

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
