# Peer review of "Clinical and Radiographic Outcomes of Inversed Restricted Kinematic Alignment Total Knee Arthroplasty by Asia Specific (Huang’s) Phenotypes, a Prospective Pilot Study"

_jcm, 2023, doi:10.3390/jcm12062110_

Round 1

Reviewer 1 Report

- line 93-94: 'the tibial cut was aimed at perpendicular to the tibia mechanical axis,...': riKA is about reconstructing the native tibial joint line obliquity, so not perpendicular to the mechanical axis. Please adjust.

- figure 4: I must say the orginal figure of this is from Charles Riviere. This is very recognizable. You can't use this figure so I suggest to leave it out of the article.

Author Response

Comments and Suggestions for Authors

- line 93-94: 'the tibial cut was aimed at perpendicular to the tibia mechanical axis,...': riKA is about reconstructing the native tibial joint line obliquity, so not perpendicular to the mechanical axis. Please adjust.

Author Response:
Thank you very much for your comment. the sentence is modified as suggestion.

- figure 4: I must say the orginal figure of this is from Charles Riviere. This is very recognizable. You can't use this figure so I suggest to leave it out of the article.

Author Response:
Thank you very much for your comment. The new figure is made originally.

Reviewer 2 Report

Dear Authors,

thank for providing the manuscript on Clinical and radiographic Outcomes of inversed restricted Kinematic Alignment Total Knee Arthroplasty by Asia specific (Huang’s) phenotypes.

Please find detailed comments below.

Title: The study type is missing: prospective Pilot Study

Abstract: The use of abbreviations is not recommended for abstracts,

If needed every abbreviation needs to be introduced. e.g. HKAA

“Correlation be- 22 tween FJS-12 and WOMAC was good at 6th month (Pearson correlation coefficient (r) =0.74”: This is well known due to the nature of both PROMS and not the primary outcome of your study

Most knees being scheduled for TKA show varus deformity prior surgery, Why are they recommended for inverse alignment. The data does not support this conclusion.

Introduction

The introduction starts boring. Everybody is aware the first sentences.

L 39 There are not various philosophies known: There are various proven procedures, old ones and newer ones like robotic assisted, augmented reality… for alignment and diverse ligament balancing approaches also.

Please better introduces why reverse alignment might be of use.

As you state the main purpose is to determine the clinical outcome of irKA-TKA in all knees separated by Huang’s phenotypes, the sample is for sure to small!!

Methods

Prospective trials should be preregistered

IRB approval seems fine

Written informed consent should be given

Data should be available at least on request or better freely accessible

Inclusion criteria must be presented more precise or did every patient over 60 receive IrKA, what would not be correct

Usually, no references should be provided in the methods. If discussions on the method are common, they should be addressed in introduction or discussion

There is not justification for the SAMPLE SIZE provided!?

The methods are rather written like guidance for surgery than the methods for a study

Because of several outcome measures for this small sample in various groups the cumulative alpha error is to big. I would recommend going for some descriptive data and providing meand and SD for further sample size calculation in a bigger trial. The statistical approach is therefore questionable. 

Results/Discussion

The whole approach lacks comparison to gold standard alignment techniques and is therefore to be introduced as a Pilot approach on reversed alignment and not as superior technique. 

Even if there are interesting issues provided I recommend to restructure the manuscript to a pilot trials discussion benefits from and problems with this technique.

Author Response

Title: The study type is missing: prospective Pilot Study

Author Response:
Thank you very much for your comment. The title is modified as suggestion.

Abstract: The use of abbreviations is not recommended for abstracts, If needed every abbreviation needs to be introduced. e.g. HKAA

Author Response:
Thank you very much for your comment. Unnecessary abbreviations are replaced.

“Correlation between FJS-12 and WOMAC was good at 6th month (Pearson correlation coefficient (r) =0.74”: This is well known due to the nature of both PROMS and not the primary outcome of your study

Author Response:
Thank you very much for your comment. The surgical results were added in the abstract to reduce the strength of PROMS comparing. After searching all reports comparing FJS-12 and WOMAC in Pubmed, the well-known association is not seen in all reports. Besides, we tried to compare WOMAC index with newly developed traditional Chinese version of FJS-12, not the English version FJS-12. Sansone et al. [1] compared WOMAC with Italian version of FJS-12 and reported the Pearson correlation coefficient between the FJS-12 and WOMAC was 0.45 (p < 0.001) at pre-op, r = 0.46 (p < 0.001) at 6 months, and r = 0.42 (p < 0.001) 12 months. Longo et al. [2] compared another Italian version of FJS-12 and WOMAC, and stated the Pearson correlation coefficient between the FJS-12 and WOMAC was 0.47 (P = 0.001) at 1 month, r = 0.57 (P < 0.001) at 3 months, and r = 0.57 (P < 0.001) at 6 months. Hindi version of FJS-12 was compared with WOMAC at 20.8±7.4 days after operation and the Pearson correlation coefficient was 0.45 [3]. And a Japanese version of FJS-12 was also compared with WOMAC with non-specific time frame, and the Pearson correlation coefficient between FJS-12 and total WOMAC score was 0.522 [4]. Thus, although FJS-12 and WOMAC may be highly associated due to the nature of the PROMS, the association may be weakened because of language translation and culture issues. In this study, we proved that the traditional Chinese version of FJS-12 is a useful and WOMAC-comparable questionnaire restrospectively. 

[1]. Sansone V, Fennema P, Applefield RC, et al. Translation, cross-cultural adaptation, and validation of the Italian language Forgotten Joint Score-12 (FJS-12) as an outcome measure for total knee arthroplasty in an Italian population. BMC Musculoskelet Disord. 2020;21(1):23.

[2]. Longo UG, De Salvatore S, Di Naro C, et al. Unicompartmental knee arthroplasty: the Italian version of the Forgotten Joint Score-12 is valid and reliable to assess prosthesis awareness. Knee Surg Sports Traumatol Arthrosc. 2022;30(4):1250-1256.

[3]. Goyal T, Sethy SS, Paul S, Choudhury AK, Das SL. Good validity and reliability of forgotten joint score-12 in total knee arthroplasty in Hindi language for Indian population. Knee Surg Sports Traumatol Arthrosc. 2021;29(4):1150-1156.

[4]. Matsumoto M, Baba T, Homma Y, et al. Validation study of the Forgotten Joint Score-12 as a universal patient-reported outcome measure. Eur J Orthop Surg Traumatol. 2015;25(7):1141-1145

Most knees being scheduled for TKA show varus deformity prior surgery, Why are they recommended for inverse alignment. The data does not support this conclusion.

Author Response:
Thank you very much for your comment. We only included patients older than 64 years and received irKA-TKA. Many other elder patients received MA or KA-TKA. Besides, irKA-TKA is generally considered as a procedure with safe, excellent reported clinical outcome, and high satisfaction at short-term follow-up. Thus, it is reasonable to make irKA-TKA an option of TKA procedure for patients. Patients with varus knees by the definition of LDFA≥90° and MPTA<87° were grouped in Type 3 and 4. Our results showed that patients with Type 4 showed the most significant improvement in all measured angles, followed by patients in Type 2 and 3. This indicated that patients with varus knees would be the best candidates to receive irKA-TKA.

The introduction starts boring. Everybody is aware the first sentences.

Author Response:
Thank you very much for your comment. The first paragraph is modified.

L 39 There are not various philosophies known: There are various proven procedures, old ones and newer ones like robotic assisted, augmented reality… for alignment and diverse ligament balancing approaches also.

Author Response:
Thank you very much for your comment. the sentence is changed to “Various knee alignment procedures were known.”

Please better introduces why reverse alignment might be of use.

Author Response:
Thank you very much for your comment. The last paragraph of introduction is modified to strengthen the importance of irKA-TKA.

As you state the main purpose is to determine the clinical outcome of irKA-TKA in all knees separated by Huang’s phenotypes, the sample is for sure to small!!

Author Response:
Thank you very much for your comment. The small sampling pool is stated in the limitation section.

Prospective trials should be preregistered. IRB approval seems fine. Written informed consent should be given

Author Response:
Thank you very much for your comment. the ethnics section is modified for better presentation.

Data should be available at least on request or better freely accessible

Author Response:
Thank you very much for your comment. the statement is changed to “The data is available by requesting.”

Inclusion criteria must be presented more precise or did every patient over 60 receive IrKA, what would not be correct

Author Response:
Thank you very much for your comment. The sentence of inclusion criteria is changed to “Patients older than 60 years old and received inversed restricted kinematic alignment (irKA) total knee arthroplasty (TKA) between January 2018 and June 2020 in the China Medical University Hospital were included for this study.”

Usually, no references should be provided in the methods. If discussions on the method are common, they should be addressed in introduction or discussion

Author Response:
Thank you very much for your comment. Only reference 14, 15 are left in the methods. They are used to suggest the method of angle measurement is the same as other studies.

There is not justification for the SAMPLE SIZE provided!?

Author Response:
Thank you very much for your comment. We added the description of sample size in the “Methods-patient selection” section as below:

“Methods

Patient selection

We use G*Power 3.1.9.7 to calculate the sample size with effect size=0.3, type I error=0.5 and power=0.8 according to statistical test: “Mean: Difference between two dependent means (matched pairs)”. Needed 90 sample size in the study. Patients older than 60 years old, received inversed restricted kinematic alignment (irKA) total knee arthroplasty (TKA) between January 2018 and June 2020 in the China Medical University Hospital and followed at least 1 year were included for this study.”

We also added the primary results of total sample in Table 2 and the corresponding description in the Result.

The methods are rather written like guidance for surgery than the methods for a study

Author Response:
Thank you very much for your comment. Since the irKA-TKA is still a new technique and there is not much reference for it, we think it is necessary to include all the surgery steps in this report for possible readers. Although this does make the article look like a guidance, this should make the whole content and conclusion easy for future replication and confirmation.

Because of several outcome measures for this small sample in various groups the cumulative alpha error is to big. I would recommend going for some descriptive data and providing meand and SD for further sample size calculation in a bigger trial. The statistical approach is therefore questionable.

Author Response:
Thank you very much for your comment. This is indeed a study with a relatively small sample size. The sample size is mainly the patients who operated on our team during this period. This is our limitation, but it is very reasonable to explain our statistics

The whole approach lacks comparison to gold standard alignment techniques and is therefore to be introduced as a Pilot approach on reversed alignment and not as superior technique.

Author Response:
Thank you very much for your comment. The pilot study is shown in the title.

Even if there are interesting issues provided I recommend to restructure the manuscript to a pilot trials discussion benefits from and problems with this technique.

Author Response:
Thank you very much for your comment. The new title shows this is a pilot retrospective study.

Reviewer 3 Report

Dear Authors,

I am pleased to submit my review of your article.

The topic is interesting, current, and relevant to our clinical practice, given the increasing number of knee replacements and the need for implants that can last a long time. Minor revisions are needed to make the article suitable for publication. 

In detail: 

Introduction: 

Line 29: "the implants [2]." You could add this relevant article: 10.1016/j.arth.2018.02.026

Line 45: "of neutral [6,7]." You could add this recent article: 10.3390/app122111085

Line 48: "for excellency [8]." You could add this relevant article: 10.1016/j.artd.2022.101090

Methods:

A minimum follow-up of patients undergoing total knee replacement is not reported. Please clarify this point.

Results:

The results should be shorter. Several concepts are reported in both text and table, making the discourse complex to read. Make the text more fluent by reporting all numerical data in the tables. Also, the tables should all be brought into the same decimal. 

Were encountered no complications? 

Discussion:

Line 239: "lateral femur in rKA [18]." Consider adding this recent article: 10.1016/j.jor.2022.06.014

Line 254- 260: "The traditional... after joint arthroplasty." Summarize the reported concept. 

Line 286: "To reach this goal [29]." You might add this recent article: 10.3390/jcm11216569

Author Response

Reviewer 3

Line 29: "the implants [2]." You could add this relevant article: 10.1016/j.arth.2018.02.026

Author Response:
Thank you very much for your comment. The suggested reference is added.

Line 45: "of neutral [6,7]." You could add this recent article: 10.3390/app122111085

Author Response:
Thank you very much for your comment. The suggested reference is added.

Line 48: "for excellency [8]." You could add this relevant article: 10.1016/j.artd.2022.101090

Author Response:
Thank you very much for your comment. The suggested reference is added.

Methods:

A minimum follow-up of patients undergoing total knee replacement is not reported. Please clarify this point.

Author Response:
Thank you very much for your comment. All included subjects were followed more than 1 year. Losing follow-up within 1 year is added in the inclusion criteria for clarity.

Results:

The results should be shorter. Several concepts are reported in both text and table, making the discourse complex to read. Make the text more fluent by reporting all numerical data in the tables. Also, the tables should all be brought into the same decimal.

Author Response:
Thank you very much for your comment. The result section is modified for easy reading and the numbers in the Table are changed except the r value in the Table 3.

Were encountered no complications?

Author Response:
Thank you very much for your comment. Yes, no complications were encountered for all the studied subjects.

Discussion:

Line 239: "lateral femur in rKA [18]." Consider adding this recent article: 10.1016/j.jor.2022.06.014

Author Response:
Thank you very much for your comment. The suggested reference is added.

Line 254- 260: "The traditional... after joint arthroplasty." Summarize the reported concept.

Author Response:
Thank you very much for your comment. The traditional Chinese version of FJS-12 is introduced in this study.

Line 286: "To reach this goal [29]." You might add this recent article: 10.3390/jcm11216569

Author Response:
Thank you very much for your comment. The suggested reference is added.

Round 2

Reviewer 2 Report

Thank you for submitting revised version